# Endothelial Progenitor Cells as a Marker of Vascular Damage But not a Predictor in Acute Microangiopathy-Associated Stroke

**DOI:** 10.3390/jcm9072248

**Published:** 2020-07-15

**Authors:** Adam Wiśniewski, Joanna Boinska, Katarzyna Ziołkowska, Adam Lemanowicz, Karolina Filipska, Zbigniew Serafin, Robert Ślusarz, Danuta Rość, Grzegorz Kozera

**Affiliations:** 1Department of Neurology, Collegium Medicum in Bydgoszcz, Nicolaus Copernicus University in Toruń, Skłodowskiej 9 Street, 85-094 Bydgoszcz, Poland; 2Department of Pathophysiology, Collegium Medicum in Bydgoszcz, Nicolaus Copernicus University in Toruń, Skłodowskiej 9 Street, 85-094 Bydgoszcz, Poland; joanna_boinska@cm.umk.pl (J.B.); katarzyna_stankowska@cm.umk.pl (K.Z.); drosc@cm.umk.pl (D.R.); 3Department of Radiology, Collegium Medicum in Bydgoszcz, Nicolaus Copernicus University in Toruń, Skłodowskiej 9 Street, 85-094 Bydgoszcz, Poland; adam.lemanowicz@cm.umk.pl (A.L.); serafin@cm.umk.pl (Z.S.); 4Department of Neurological and Neurosurgical Nursing, Collegium Medicum in Bydgoszcz, Nicolaus Copernicus University in Toruń, Łukasiewicza 1 Street, 85-821 Bydgoszcz, Poland; karolinafilipskakf@gmail.com (K.F.); robert_slu_cmumk@wp.pl (R.Ś.); 5Medical Simulation Centre, Medical University of Gdańsk, Faculty of Medicine, Dębowa 17 Street, 80-208 Gdańsk, Poland; gkozera1@wp.pl

**Keywords:** endothelial progenitor cells, ischemic stroke, hemorrhage, prognosis, clinical outcome

## Abstract

Background: The aim of the study was to assess the number of endothelial progenitor cells (EPCs) in patients with acute stroke due to cerebral microangiopathy and evaluate whether there is a relationship between their number and clinical status, radiological findings, risk factors, selected biochemical parameters, and prognosis, both in ischemic and hemorrhagic stroke. Methods: In total, 66 patients with lacunar ischemic stroke, 38 patients with typical location hemorrhagic stroke, and 22 subjects from the control group without acute cerebrovascular incidents were included in the prospective observational study. The number of EPCs was determined in serum on the first and eighth day after stroke onset using flow cytometry and identified with the immune-phenotype classification determinant (CD)45−, CD34+, CD133+. Results: We demonstrated a significantly higher number of EPCs on the first day of stroke compared to the control group (med. 17.75 cells/µL (0–488 cells/µL) vs. 5.24 cells/µL (0–95 cells/µL); *p* = 0.0006). We did not find a relationship between the number of EPCs in the acute phase of stroke and the biochemical parameters, vascular risk factors, or clinical condition. In females, the higher number of EPCs on the first day of stroke is related to a favorable functional outcome on the eighth day after the stroke onset compared to males (*p* = 0.0355). We found that a higher volume of the hemorrhagic focus on the first day was correlated with a lower number of EPCs on the first day (correlation coefficient (R) = −0.3378, *p* = 0.0471), and a higher number of EPCs on the first day of the hemorrhagic stroke was correlated with a lower degree of regression of the hemorrhagic focus (R = −0.3896, *p* = 0.0367). Conclusion: The study showed that endothelial progenitor cells are an early marker in acute microangiopathy-associated stroke regardless of etiology and may affect the radiological findings in hemorrhagic stroke. Nevertheless, their prognostic value remains doubtful in stroke patients.

## 1. Introduction

Stroke is an important social and medical problem in the 21st century, as it is one of the main causes of morbidity and long-term disability and the second most frequent cause of death in the world [1]. Ischemic stroke associated with disturbances of the blood flow to the brain tissue, leading to necrosis of the part of the brain covered by ischemia (80–85%), is the most common. Hemorrhagic stroke (15–20%) associated with extravasation of blood to the brain is less common but has greater mortality [2]. Stroke may be the result of endothelial dysfunction (in the course of cerebral microangiopathy), as well as the cause of vascular endothelial damage. Stroke due to small vessel disease, i.e., lacunar stroke, is associated with pathological changes (classical atherosclerosis, fibrosis, enamel, and calcification) of small cerebral vessels (diameter below 600 µm), and accounts for approximately 20–25% of all ischemic strokes [3]. Among hemorrhagic strokes, the most common, typical location (deep) intracerebral hemorrhage, seems to have been most related to microangiopathy. It results from blood extravasation from stabbing branches (most often lenticular-striatum arteries) supplying the basal ganglia and thalamus. It is related to pathological changes of the vessel’s walls (including cells of the endothelium) in the course of improperly treated hypertension [4].

Endothelial progenitor cells (EPCs) are a recognized marker of both the degree of endothelial damage and the ability to regenerate the endothelium. Due to their multiplication potential, they can differentiate into many cells, but most often, they proliferate into mature circulating endothelial cells. Thanks to mediators, such as VEGF (vascular endothelial growth factor), SDF-1 (stromal-derived factor), and G-CSF (granulocyte colony-stimulating factor), they migrate to damaged areas of the brain affected by ischemia. They play an important role in the regeneration of the nervous tissue, glial cell nutrition, reduction of neuronal apoptosis, and blood–brain barrier stabilization. They are associated with postnatal angiogenesis, especially in neovascularization of blood vessels damaged by ischemia [5,6,7]. This is particularly important in patients with stroke. As a result, they are increasingly considered as a potential treatment method for stroke patients, especially with ischemic stroke, and the initial results of their use in studies on mice and rats seem encouraging [8,9,10,11,12]. However, the role and importance of these cells in stroke patients is still the subject of controversy, and reports on this subject are scarce and often ambiguous. It is believed that a large number of EPCs, due to their regenerative and repairing properties, may affect the size of the ischemic focus and even the clinical and functional status of the patients, and thus, the prognosis [13]. Therefore, the aim of this study was to assess the number of EPCs in the blood serum and their potential relationship with the clinical condition, radiological image, and prognosis in patients in the acute phase of stroke caused by cerebral microangiopathy.

## 2. Material and Methods

### 2.1. Study Design and Participants

The study was conducted in accordance with the Declaration of Helsinki and the protocol was approved by the Bioethics Committee of Nicolaus Copernicus University in Torun at Collegium Medicum of Ludwik Rydygier in Bydgoszcz (KB No. 769/2014). The study included subjects who, having read the study protocol, signed the informed consent to participate in the study. The researchers explained all stages of the study to the subjects and presented all potential risks associated with the research.

The definition of stroke, updated in 2013 by the American Heart Association/American Stroke Association (AHA/ASA), was used, which is an episode of a sudden neurological disorder caused by focal cerebral, spinal, or retinal ischemia lasting over 24 h or corresponding to the morphological features of ischemia of the central nervous system [14].

The study was conducted from February 2015 to December 2017 in the Department of Neurology at Collegium Medicum in Bydgoszcz of Nicolaus Copernicus University in Torun in the University Hospital No. 1 in Bydgoszcz. This prospective and observational study included stroke patients with cerebral microangiopathy: 66 patients with lacunar ischemic stroke, 38 patients with typical location intracerebral hemorrhage, and 22 people from the control group.

The group of patients with lacunar stroke included patients with no significant hemodynamic stenoses of large pre-skull vessels or cardiogenic-embolic background, and the performed neuroimaging confirmed the presence of a lacunar focus and/or revealed chronic vascular changes with a typical location and morphology (subcortical lesions, periventricular lesions, leukoaraiosis features) [15]. The typical location intracerebral hemorrhage was diagnosed based on the results of a computed tomography scan performed during the patient’s admission. Patients with symptomatic cerebral hemorrhage in the course of taking oral anticoagulants and patients with extensive lobal hemorrhage during amyloid angiopathy were excluded. We included only stroke subjects admitted to the hospital with a duration of stroke symptoms no longer than 12 h. The control group consisted of people of similar age and vascular risk factors hospitalized in the Department of Neurology for reasons other than acute cerebrovascular disease and did not represent cerebrovascular incidents in the last 3 years.

Exclusion criteria included lack of the patient’s consent to participate in the study or inability to express it consciously (stroke with aphasia or quantitative disturbances of consciousness); patients with documented oncological history; patients with chronic inflammatory processes, e.g., chronic venous thrombosis of the lower limbs or chronic ischemia of the lower limbs; patients with a stroke or TIA during the last 3 years; and patients with severe bleeding in the last 2 years, e.g., gastrointestinal bleeding, level of hemoglobin < 9 g/dL, hematocrit value < 35%; and duration of stroke symptoms more than 12 h before hospital admission.

Routine laboratory tests were performed at the Laboratory Diagnostics Department of the University Hospital No. 1 in Bydgoszcz in the morning within 24 h from the onset of stroke symptoms. About 6 mL of blood were collected from the veins of the forearm from patients to determine the following biochemical parameters in the blood serum: C reactive protein (CRP), fibrinogen, and homocysteine (Atellica Solution, Siemens Healthcare, Erlangen, Germany).

In all patients, computed tomography without contrast was performed at the time of admission to the hospital in the Hospital Emergency Department of A. Jurasz University Hospital No. 1 in Bydgoszcz using a 64-row Brilliance computer CT scanner (Phillips, Eindhoven, The Netherlands). In subjects with hemorrhagic stroke, the volume of hemorrhagic focus was assessed in mL on the 1st and 8th day of the stroke using the special Philips software. The degree of hemorrhagic focus regression was the volume difference between the 1st and 8th day.

### 2.2. Flow Cytometry

Determination of EPCs in blood in stroke subjects was performed on admission, within the first 24 h (1st day), and on the 8th day of the disease, using flow cytometry. In the control group, cell determinations were made on the 1st day of the hospital stay. The method for the determination of the level of circulating EPCs was based on previous reports [16,17]. Fresh blood (4.5 mL) with minimal stasis was collected into cooled tubes (Becton Dickinson Vacutainer^®^ System, Plymouth, UK) containing potassium ethylenediaminetetraacetic acid (K2EDTA) and analyzed within 2 h. The samples were obtained in the morning between 8 and 10 a.m., after a 12-h period of overnight fasting. The approach of the current study was to use three concurrent markers of classification determinant (CD)45−, CD34+, CD133+, to increase the accuracy of endothelial progenitor detection. Cells were further confirmed by a fluorescent-activated cell sorting (FACS) Calibur flow cytometer (Becton Dickinson, San Diego, USA) using monoclonal antibodies directed against antigens specific for circulating endothelial progenitor cells (Figure 1). The data acquired was analyzed by using CellQuest software (Becton Dickinson). Circulating EPC counts were assessed by flow cytometry according to the procedure provided by Mancuso et al. [16]. Fresh peripheral blood (50 μL) was incubated with Peridinin-Chlorophyll-Protein–Cyanine (PerCP-Cy5.5)-conjugated anti-CD45 (concentration 25 µg/mL), as well as allophycocyanin (APC)-conjugated anti-CD34 antibodies (concentration 25 µg/mL) (all BD Biosciences, Pharmingen, San Diego, CA, USA), and phycoerythrin (PE)-conjugated anti-CD133 (concentration 50 µg/mL) (Miltenyi Biotec, Bergisch Gladbach, Germany). EPCs were defined as negative for hematopoietic marker CD45, positive for endothelial progenitor marker CD133, and positive for endothelial cell marker CD34, showing expression on early hematopoietic and vascular-associated tissue. At least 100,000 events were measured in each sample. The total cell count was calculated by TruCount tubes (BD Biosciences, San Jose, CA, USA) containing a calibrated number of fluorescent beads, and ‘lyse-no-wash’ procedures were used in the present study to improve the sensitivity [17]. Absolute EPCs numbers (cells/μL) were calculated based on the following pattern: Number of measured EPCs/number of fluorescent beads counted × number of beads/μL.

### 2.3. Clinical Outcome

Both the clinical and functional condition were assessed by means of standardized research tools: the National Institute of Health Stroke Scale (NIHSS) and the Modified Rankin Scale (mRS) [18,19], within the first 24 h after admission to the hospital (1st day) and on the 8th day of hospitalization. Two subgroups of stroke patients were identified based on the stroke severity: A subgroup with a mild and moderate neurological deficit (0–10 points on the NIHSS scale), and a subgroup with a severe neurological deficit (>10 points on the NIHSS scale). Due to the functional condition, two subgroups of patients with stroke were identified: Those with a favorable prognosis (0–2 points on the mRS scale) and those with an unfavorable prognosis (3–5 points on the mRS scale).

### 2.4. Statistical Analysis

The statistical analysis of collected data was performed with the help of the statistical program STATISTICA—version 13.1 (Dell Inc., Round Rock, TX, USA). Due to the unfulfilled assumptions related to the possibility of using parametric tests (Shapiro–Wilk for normality and Levene’s for homogeneity of variance), non-parametric tests were used in the analysis, namely the Mann–Whitney U test, Wilcoxon test, Kruskal–Wallis test, Spearman’s rank correlation test, and independence chi-square test. Variables not characterized by normal distribution were described using the median (median value), quartile distribution, and range. Multivariate regression analysis (MANOVA) was conducted to estimate relations between EPCs and clinical or functional condition. The significance level *p* < 0.05 was considered statistically significant.

## 3. Results

The general characteristics and comparison of the population of the studied patients are presented in Table 1. Patients with hemorrhagic stroke were in a significantly worse functional condition (mRS) on the first day compared to ischemic stroke subjects.

There was a significantly higher number of EPCs in the blood on the first day of stroke (regardless of etiology) compared to the control group (respectively, med. 17.75 cells/µL (0–488 cells/µL) vs. 5.24 cells/µL (0–95 cells/µL); *p* = 0.0006). There was a significantly higher number of EPCs in the blood serum on the first day of ischemic stroke compared to the control group (med. 18.65 cells/µL (0–278 cells/µL) vs. 5.24 cells/µL (0–95 cells/µL); *p* = 0.0011) and on the first day of hemorrhagic stroke compared to the control group (med. 17.17 cells/µL (0–488 cells/µL) vs. 5.24 cells/µL (0–95 cells/µL); *p* = 0.0034) (Figure 2).

There were no significant differences in the number of EPCs between patients with ischemic and hemorrhagic stroke both on the first day and on the eighth day of the disease. Prospective analysis did not show significant changes in the number of EPCs in the blood of patients with stroke (regardless of etiology) between the first and eighth day of the disease.

The number of EPCs did not differ significantly in men and women, both in the whole population and in the group with stroke (regardless of etiology) on the first day or in the group with stroke on the eight day of the disease. There were no significant correlations between the age and the number of EPCs on the first day in the whole population (R = 0.0370, *p* = 0.6809), and in patients with hemorrhagic stroke on the first (R = 0.0783, *p* = 0.6402) and eighth day (R = −0.0762, *p* = 0.6489) and ischemic stroke on the first (R = −0.0326, *p* = 0.7949) and eighth day (R = 0.0939, *p* = 0.4529).

There were no significant relationships between the number of EPCs on the first and eighth day after a stroke event with hypertension, hyperlipidemia, smoking, and diabetes in ischemic stroke (Table 2), as well as in hemorrhagic stroke (Table 3).

There were no significant correlations between EPCs on the first and eighth day after a stroke event with the selected biochemical parameters, both in ischemic and hemorrhagic stroke (Table 4).

There were no significant correlations between the number of EPCs on the first day of stroke (regardless of etiology) and the clinical condition (NIHSS scale) on the first day (R = 0.0128; *p* = 0.8790) and on the eighth day (R = 0.1300; *p* = 0.1882), as well as between the number of EPCs on the eighth day of stroke and the clinical condition on the first day (R = 0.1846; *p* = 0.0607) and on the eighth day (R = 0.1243; *p* = 0.2085). There were no significant relationships between the number of EPCs on the first day of stroke (regardless of etiology) and the functional condition (mRS scale) on the first day (R = 0.0318; *p* = 0.7480), and on the eighth day (R = −0.1239; *p* = 0.2099), as well as between the number of EPCs on the eighth day of stroke and the functional condition on the first day (R = 0.0049; *p* = 0.9606) and on the eighth day (R = 0.0672; *p* = 0.4973). Considering the etiology of stroke, there were no significant correlations between the number of EPCs on the first and eighth day of the disease and the clinical or functional condition, both in ischemic and hemorrhagic stroke (Table 5).

There were no significant differences between patients with severe and mild neurological deficit on the first day of stroke in relation to the number of EPCs on the first day (total *p* = 0.4802; ischemic stroke *p* = 0.7837; hemorrhagic stroke *p* = 0.4166) and on the eighth day (total *p* = 0.1794; ischemic stroke *p* = 0.2969; hemorrhagic stroke *p* = 0.4457). Similarly, there were no significant differences between patients with severe and mild neurological deficit on the eighth day of stroke in relation to the number of EPCs on the first day (total *p* = 0.4545; ischemic stroke *p* = 0.3248; hemorrhagic stroke *p* = 0.9568) and on the eighth day (total *p* = 0.6479; ischemic stroke *p* = 0.2069; hemorrhagic stroke *p* = 0.5335). There were no significant differences between patients with favorable and unfavorable prognosis on the first day of stroke in relation to the number of EPCs on the first day (total *p* = 0.9383; ischemic stroke *p* = 0.8786; hemorrhagic stroke *p* = 0.8903) and on the eighth day (total *p* = 0.9072; ischemic stroke *p* = 0.9264; hemorrhagic stroke *p* = 0.9278). Similarly, there were no significant differences between patients with a favorable and unfavorable prognosis on the eighth day of stroke in relation to the number of EPCs on the first day (total *p* = 0.1470; ischemic stroke *p* = 0.2369; hemorrhagic stroke *p* = 0.4559) and on the eighth day (total *p* = 0.6969; ischemic stroke p = 0.9485; hemorrhagic stroke *p* = 0.4559).

In the multivariate model of regression adjusted for sex, type of stroke, and clinical or functional condition, we demonstrated that in females, a higher number of EPCs on the first day of stroke is related to a favorable outcome on the eighth day after the stroke onset compared to males (*p* = 0.0355) (Figure 3). There were no significant correlations regarding the other analyzed dependencies.

There was a negative but significant correlation between the volume of hemorrhagic focus on the first day of hemorrhage and the number of EPCs on the first day of hemorrhagic stroke (R = −0.3378, *p* = 0.0471) (Figure 4).

There were no significant correlations between the number of EPCs on the eighth day of hemorrhagic stroke with the volume of hemorrhagic focus on the first day (R = −0.0791, *p* = 0.6513) and on the eighth day (R = −0.0002, *p* = 0.9897), as well as between the number of EPCs on the first day and the volume of hemorrhagic focus on the eighth day (R = −0.1294, *p* = 0.4803). A negative correlation between the number of EPCs on the first day of hemorrhagic stroke and the degree of regression of the hemorrhagic focus was demonstrated (R = −0.3896, *p* = 0.0367) (Figure 5).

## 4. Discussion

The results of this study showed that, in the acute phase of ischemic and hemorrhagic stroke, a significantly higher number of EPCs were observed than in the control group. This confirms that damage to the cerebral endothelium, whether in the course of acute ischemia or mechanical damage to the vascular wall, leads to significant mobilization and proliferation of EPCs. This is probably the mechanism of differentiation into mature endothelial cells, which, due to the production of numerous cytokines, are actively involved in the repair and neovascularization of damaged brain tissues [7]. Similar conclusions were drawn by Yip et al., Meamar et al., and Regueiro et al. [20,21,22], who, assessing patients in the acute phase of ischemic stroke, also found a significantly higher number of EPCs than in control. Similarly, Paczkowska et al. [23] obtained a higher number of EPCs in the acute stage of hemorrhagic stroke in comparison to the control group. The results of our research and the above show that regardless of the etiology, it is the state of sudden damage to vascular endothelium (similar to acute myocardial infarction and acute limb ischemia) that clearly activates EPCs’ proliferation, where EPCs are the main repair and regenerative element of the damaged endothelial cells [24,25]. It is worth noting that Ghani et al. and Deng Y et al. [26,27] in their studies obtained different results and a lower number of EPCs in the acute phase of ischemic stroke than those in the control group, and Zhou et al. [28] noted a lower number of EPCs in patients with both acute ischemic and hemorrhagic stroke, compared to the control group. The differences in the results of the above studies could have resulted from a different population of patients and a different configuration of superficial antigens used in flow cytometry to detect EPCs.

The data in the literature show that, in acute cerebrovascular incidents, EPCs are activated within the first 24 h, reach their maximum blood level around the 7th day, and gradually decrease after 21–28 days. Most authors assessed only patients with ischemic stroke [27,29,30] and only a few assessed patients with either ischemic or hemorrhagic stroke [28]. Taguchi et al. and Marti-Fabregas et al. [29,30] showed statistically significantly more EPCs after 7 days of acute ischemic stroke than during the first 24 h. Zhou et al. [28] also obtained similar results but in both ischemic and hemorrhagic stroke. In this work, although in both types of stroke the number of cells was higher on the eighth day of the disease than on the first day, these differences did not reach statistical significance. Nevertheless, the results of this study confirmed that mobilization of EPCs occurs within a few hours after the onset of symptoms of stroke; within 24 h, the number of EPCs in the blood serum reaches a very high level, and the state of high activation persists for at least the first week of the disease. Due to the fact that the area of interest was the acute phase of cerebrovascular incidents, the number of EPCs on day 21–28 was not assessed. In addition, in this study, the number of EPCs did not differ significantly in patients with ischemic and hemorrhagic stroke (similar results were presented by Zhou et al.), both on the first and on the eighth day, which suggests that acute brain endothelial damage, and not its etiopathogenesis, plays a leading role in the activation of EPCs.

In this research, no significant correlations were found between the number of EPCs and the selected biochemical parameters of blood (CRP, homocysteine, fibrinogen), as well as risk factors of vascular diseases, which suggests their potential lack of influence on the number of activated EPCs. It is suggested that the above factors may affect the chronic number of EPCs in the blood, without affecting their mobilization and activation capacity in acute endothelial damage, such as in stroke.

In the present study, it was not shown that the number of EPCs in the acute phase of stroke significantly affected the clinical and functional status of the patients or was associated with early prognosis. There were no significant correlations between the number of EPCs on the first and eighth day of the stroke and the score on the mRS or NIHSS scale on the first and eighth day, both in ischemic and hemorrhagic stroke. In addition, the division of patients into groups with a favorable and unfavorable prognosis and with a mild and severe neurological deficit did not differentiate both types of strokes based on the number of EPCs. Zhou et al. also did not demonstrate the effect of the number of EPCs on the clinical state and prognosis of patients (both in ischemic and hemorrhagic stroke) and Marti-Fabregas et al. did not find a significant correlation in ischemic stroke [28,30]. In contrast, Sobrino et al. [31] noted that a large number of EPCs on the seventh day of ischemic stroke is associated with a better clinical condition, expressed by a lower score of the NIHSS scale. However, it should be noted that they were only evaluating patients with non-lacunar stroke, while the analysis of this work is related only to patients with lacunar stroke. On the other hand, Yip et al. [20] noted that a small number of EPCs on the second day of ischemic stroke is associated with a worse clinical deficit, expressed by a higher score on the NIHSS scale. However, they took into account all patients with ischemic stroke, regardless of the etiopathogenesis, i.e., both patients with lacunar and non-lacunar strokes. Pias-Peleteiro et al. [32] analyzed the relationships between the number of EPCs and the functional condition and prognosis in patients with hemorrhagic stroke and showed that a higher number of EPCs on the seventh day is associated with a better distant prognosis expressed by a small number of points on the mRS scale on the 12th month from the hemorrhage. Conversely, Sobrino et al. [33] noted that a large number of EPCs on the seventh day of hemorrhagic stroke is associated with a better prognosis and functional state of patients in the third month from the hemorrhage, also expressed by lower scores on the mRS scale. Nevertheless, ambiguous and often contradictory results of studies on the impact of EPCs on the prognosis of stroke patients suggest further research in this subject.

Multivariate analysis showed that the relation between the number of EPCs and functional condition may depend on the sex. In females, a higher number of EPCs is related with a favorable functional status. This preliminary finding reported in this study underlines the potential impact of sex hormones for a possible role of EPCs in stroke prognosis. More research is required to improve these initial findings.

The results of this study showed that a large number of EPCs on the first day of hemorrhagic stroke is associated with a smaller volume of the hemorrhagic focus. Zhou et al. [26] also analyzed similar relationships but found no significant association between the number of EPCs in the acute stage of hemorrhagic stroke and the volume of the hemorrhagic focus. In contrast, Pias-Peleteiro et al. and Sobrino et al. [32,33] showed a similar significant negative correlation between the number of EPCs on the seventh day of hemorrhagic stroke and the volume of the residual hemorrhagic focus, respectively, on the third and sixth month after the hemorrhage. Other authors analyzed the volume of the ischemic focus and showed that a large number of EPCs in the acute phase of ischemic stroke is associated with a relatively smaller volume of the ischemic focus in the diffusion sequence (DWI) [31,34]. In the present study, the relationship between the number of EPCs and the volume of the ischemic focus was not analyzed.

To our best knowledge, this is the first study analyzing the relationship between the number of EPCs and the degree of regression of the hemorrhagic focus. The significant negative correlation obtained in this research is a novelty in this field and is in contradiction to the well-known repair and regenerative function of EPCs suggested by most authors. Subjects with a higher number of EPCs on the first day presented with the lowest regression level of the hematoma volume. The results reported in this study may shed new light on the role of EPCs in hemorrhagic stroke and significantly undermine and raise doubt to their repair properties. Most of the recent pre-clinical studies in animal models demonstrated a protective and regenerative function of EPCs after cerebrovascular insult [12,35,36,37,38]. Several mechanisms of possible action were reported, especially by suppressing oxidative stress, apoptosis, mitochondrial impairment, and inflammation processes [35,36,37]. The essential role of EPCs in increasing brain angiogenesis has been highlighted in animal models of stroke [12,38]. However, the lack of references in the literature and the small number of patients with hemorrhagic stroke in this study suggest that verification of the obtained data and further research in this area in multi-center randomized trials is needed.

This study has its limitations. The effect of the number of EPCs on the distant prognosis in patients with stroke was not analyzed. The moderate number of patients and small control group is also a limitation. However, these numbers seemed sufficient to draw conclusions. Conversely, for formal reasons (conscious consent for the study for the bioethics committee), the analysis did not include patients with a severe neurological deficit, e.g., patients with consciousness disorders, so the study did not include the actual cross-section of patients with stroke but only patients with a milder clinical condition. Determination of EPCs by only one measurement at different times within the first 24 h of stroke is also a main limitation and could have a great impact on the results. The authors are aware that for the dynamic changes in the function and number of EPCs under ischemic or inflammatory conditions, the use of microbeads and Q-dot-based nanoparticles is superior to conventional flow cytometry. Most statistical analyses were univariate, which could have reduced the reliability of the results.

## 5. Conclusions

The study showed that endothelial progenitor cells are an early marker of cerebral vascular damage, both in ischemic and hemorrhagic stroke. The research highlights, for the first time, a negative correlation between the level of EPCs and the degree of regression of a hemorrhagic focus and that this relation between the number of EPCs and functional condition may depend on the sex. However, the prognostic value of EPCs for the clinical condition and early prognosis of stroke patients remains doubtful.

## Figures and Tables

**Figure 1 jcm-09-02248-f001:**
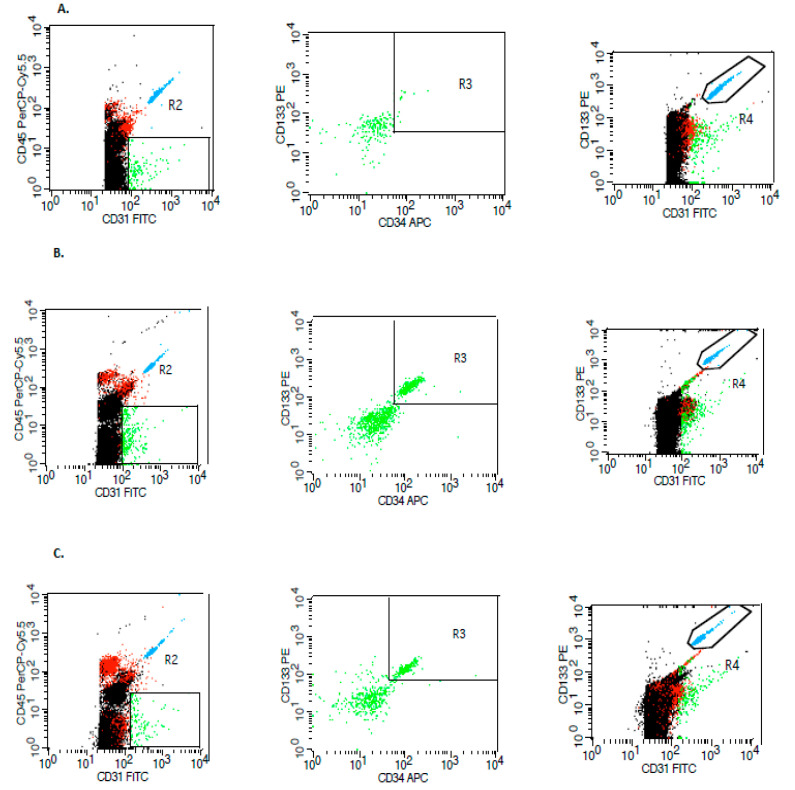
Sample of selected flow cytometric plots for the identification of circulating endothelial. progenitor cells in the control group (**A**), in patients with ischemic stroke (**B**), and hemorrhagic stroke (**C**). Peridinin-Chlorophyll-Protein–Cyanine-conjugated anti-CD45 (CD45 PerCP-Cy5.5), allophycocyanin-conjugated anti-CD34 antibodies (CD34 APC), phycoerythrin-conjugated anti-CD133 (CD133 PE), fluorescin isothiocyanate anti-CD31 (CD31 FITC). R2,R3,R4—regions defined in flow-cytometric dot plots for the detection of relevant surface markers of mononuclear cells, R2—gate for CD45 PerCP-Cy5.5/CD31 FITC; R3—gate for CD133 PE/CD34 APC; R4—gate for CD133 PE/CD31 FITC.

**Figure 2 jcm-09-02248-f002:**
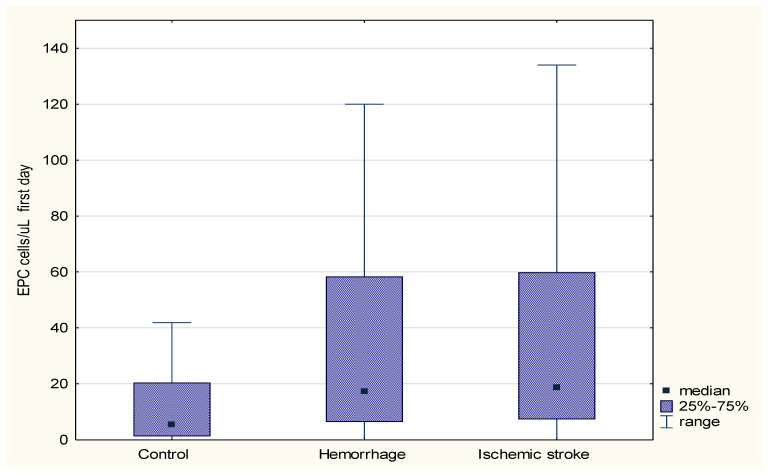
Comparison of the number of endothelial progenitor cells (EPCs) on the first day between the patients with ischemic stroke, hemorrhagic stroke, and the control group.

**Figure 3 jcm-09-02248-f003:**
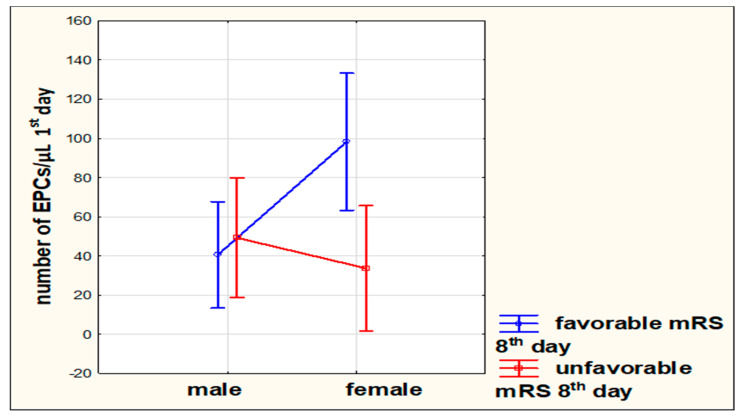
Multivariate analysis between the number of endothelial progenitor cells (EPCs) on the first day of stroke, sex, and functional outcome on the eighth day in the modified Rankin scale (mRS).

**Figure 4 jcm-09-02248-f004:**
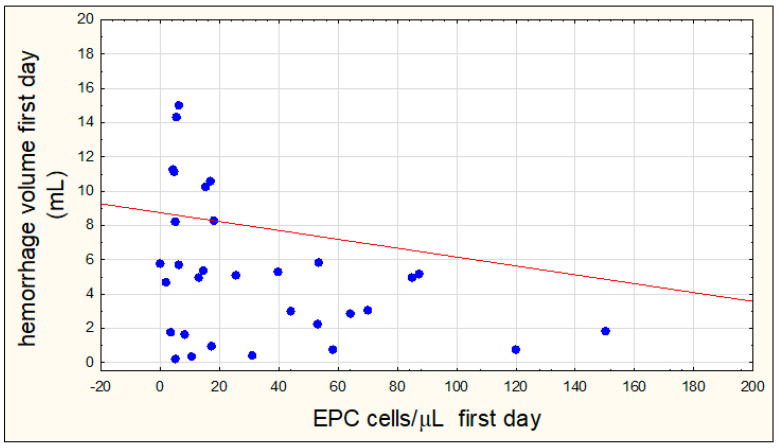
Correlation between the number of endothelial progenitor cells (EPCs) on the first day of hemorrhagic stroke and the volume of hemorrhagic focus on the first day of stroke.

**Figure 5 jcm-09-02248-f005:**
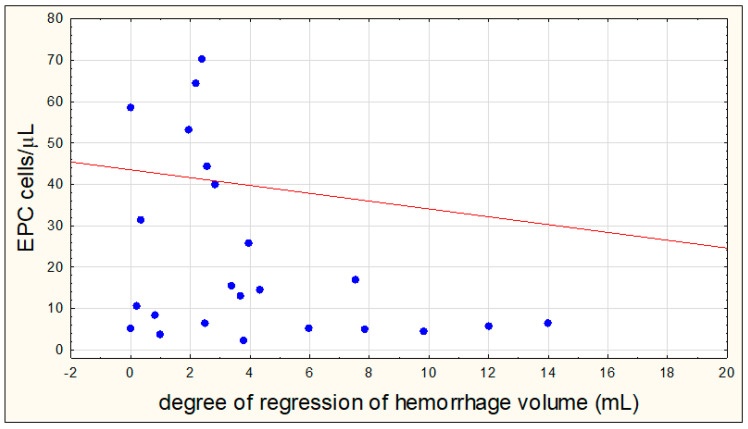
Correlation between the number of endothelial progenitor cells (EPCs) on the first day of hemorrhagic stroke and the degree of regression of the hemorrhagic focus.

**Table 1 jcm-09-02248-t001:** Comparison of selected anthropometric, biochemical parameters, risk factors, and clinical status in patients with ischemic stroke, hemorrhagic stroke, and in the control group.

Parameter	Ischemic Stroke	Hemorrhage	Control Group	*p*-Values
Sex, male, (%) ^1^	62.1	55.3	36.4	0.1088
Sex, female, (%) ^1^	37.9	44.7	63.6	0.1267
Age (median, range) ^3^	69 (45–88)	73.5 (45–91)	63.5 (50–82)	0.1034
Smoking, (%) ^1^	32.6	28.9	24.5	0.3457
Hypertension, (%) ^1^	90.9	94.7	81.8	0.2555
Hyperlipidemia, (%) ^1^	60.6	50	54.5	0.566
Diabetes, (%)^1^	37.9	21.8	27.3	0.186
CRP (mg/L), (median, range) ^2^	4.50 (0.39–58.12)	5.79 (0.38–70.1)	-	0.2117
Homocystein (µg/mL) (median, range) ^2^	11.05 (3.52–30.92)	9.22 (2.65–42.8)	-	0.6341
Fibrinogen (g/L), (median, range) ^2^	284 (59–590)	315.5 (157–463)	-	0.2985
NIHSS 1st day (points) (median, range) ^2^	6 (2–21)	6 (1–21)	-	0.6103
NIHSS 8th day (points) (median, range) ^2^	3 (0–15)	3 (0–14)	-	0.7086
mRS 1st day (points) (median, range) ^2^	4 (2–5)	5 (3–5)	-	0.0001 *
mRS 8th day (points) (median, range) ^2^	2 (0–5)	3 (0–4)	-	0.2377

^1^ chi square test, ^2^ Mann–Whitney U test, ^3^ Kruskal–Wallis test, * statistical significance, CRP, C-reactive protein; NIHSS, National Institute of Health Stroke Scale; mRS, modified Rankin Scale.

**Table 2 jcm-09-02248-t002:** Comparison of the number of endothelial progenitor cells (EPCs) on the first and eighth day of ischemic stroke between patients with present and absent vascular risk factors.

	EPCs/µL 1st Day			EPCs/µL 8th Day		
Parameter	Present	Absent	*p*-Values *	Present	Absent	*p*-Values *
Hypertension	17.14 (0–278.11)	23.24 (2.77–112.21)	0.8847	24.45 (0.46–316.89)	62.47 (18–323.29)	0.0536
Hyperlipidemia	14.69 (0–178.86)	25.55 (4.12–278.11)	0.1785	25.02 (0.40–316.89)	25.72 (2.03–323.29)	0.4724
Diabetes	14.47 (0–178.86)	25.12 (1.01–278.11)	0.0701	31.16 (0.40–323.29)	24.51 (0.46–316.89)	0.7014
Smoking	16.32 (0–178.86)	22.13 (2.77–112.21)	0.1654	27.85 (0.40–323.29)	31.69 (2.03–323.29)	0.1324

* Mann–Whitney U test. Results are median (range) in cells/µL.

**Table 3 jcm-09-02248-t003:** Comparison of the number of endothelial progenitor cells (EPCs) on the first and eighth day of hemorrhagic stroke between patients with present and absent vascular risk factors.

EPCs/µL 1st Day				EPCs/µL 8th Day		
Parameter	Present	Absent	*p*-Values *	Present	Absent	*p*-Values *
Hypertension	17.75 (0–488.41)	4.43 (3.75–5.11)	0.0722	17.93 (0–325.43)	22.44 (0.10–44.78)	0.4522
Hyperlipidemia	31.23 (0.62–338.00)	15.51 (0–488.41)	0,7042	37.62 (1.21–325.43)	11.64 (0–233.20)	0.1443
Diabetes	17.48 (6.36–58.38)	16.47 (0–488.41)	0.7608	17.15 (3.01–10020)	17.93 (0–325.43)	0.7608
Smoking	25.67 (0.62–338.00)	21.98 (0–488.41)	0.6983	23.68 (1.21–325.43)	17.44 (0.10–233.20)	0.6684

* Mann–Whitney U test Results are median (range) in cells/uL.

**Table 4 jcm-09-02248-t004:** Correlations between the number of endothelial progenitor cells (EPCs) on the first and eighth day of ischemic and hemorrhagic stroke and the selected biochemical parameters.

EPCs/µL 1st Day			EPCs/µL 8th Day					
	Ischemic Stroke		Hemorrhage		Ischemic Stroke		Hemorrhage	
	R	*p*	R	*p*	R	*p*	R	*p*
CRP	0.1630	0.1909	−0.0242	0.8854	0.0986	0.4308	−0.1526	0.3602
fibrinogen	−0.1731	0.1644	0.1135	0.4974	0.1095	0.3816	−0.0459	0.7840
homocystein	−0.0879	0.4827	0.0578	0.7309	−0.0376	0.7639	0.2465	0.1356

Spearman’s rank correlation, CRP, C-reactive protein, R, correlation coefficient.

**Table 5 jcm-09-02248-t005:** Correlations between the number of endothelial progenitor cells (EPCs) on the first and eighth day of ischemic and hemorrhagic stroke and the clinical and functional status on the first and eighth day of stroke.

EPCs/µL 1st Day			EPCs/µL 8th Day					
	Ischemic Stroke		Hemorrhagic Stroke		Ischemic Stroke		Hemorrhagic Stroke	
	R	*p*	R	*p*	R	*p*	R	*p*
NIHSS 1st day	−0.0469	0.7084	0.0932	0.5778	0.1842	0.1387	0.2108	0.2038
NIHSS 8th day	−0.1469	0.2388	−0.0888	0.5959	0.1446	0.2465	0.0857	0.6085
mRS 1st day	0.1359	0.2765	−0.1228	0.4624	0.0230	0.8544	0.0837	0.6171
mRS 8th day	−0.1355	0.2766	0.1300	0.1882	0.0858	0.4933	0.0648	

Spearman’s rank correlation, NIHSS, National Institute of Health Stroke Scale, mRS, modified Rankin Scale. R, correlation coefficient.

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
