# Peer review of "Endothelial Progenitor Cells as a Marker of Vascular Damage But not a Predictor in Acute Microangiopathy-Associated Stroke"

_jcm, 2020, doi:10.3390/jcm9072248_

Round 1

Reviewer 1 Report

Author revised manuscript according to the reviewer's comment and this is improved for revision.

Additionally, authors are suggested to look through manuscipt again.In case that author can not provide new data, author should explain sufficiently for readers.

Minor comment: In figure 1, CD31 FITC diagram looks strange. Dot in FACS should be spreaded but near CD31 FITC 101, dot patten may be cut (Left side). Even taking into account the negative control, it is extraordinary. There are may be difference depending on device (voltage...) Please check it.

Author Response

At the beginning, I would like to thank You for the careful review of our study and the constructive comments that have been used to organize all issues and improve the work.

Point 1:

Author revised manuscript according to the reviewer's comment and this is improved for revision.

Response:

Thank You. According to Reviewer’s comments we revised our paper very carefully, in particular we shortened a lot of  sentences  and corrected  the grammatical issues to increase a readibility of this paper.  We did our best to make this manuscript more clear and transparent. Additionally we have made extensive changes of language and structure in our manuscript,  using the MPDI professional English editing service.

Point 2:

Additionally, authors are suggested to look through manuscipt again. In case that author can not provide new data, author should explain sufficiently for readers.

Response:

It true it is difficult to demonstrate new data in this area, but please notice that we showed some new aspects in our paper. We reported, as the first, the inversely relation of regression of hematoma and EPCs number. We noticed the possible role of sex in the relationship of EPCs number and functional condition. Failure to demonstrate the significant relationships between EPCs and prognosis or clinical condition should also be considered as an important issue that denies the role of these cells in stroke recovery. Summarizing, we have shed light on several new aspects in this field and we underlined them  in our paper (Lines 367-368, 379-385).Please consider this novelties as an advantage of our study.

Point 3:

Minor comment: In figure 1, CD31 FITC diagram looks strange. Dot in FACS should be spreaded but near CD31 FITC 101, dot patten may be cut (Left side). Even taking into account the negative control, it is extraordinary. There are may be difference depending on device (voltage...) Please check it.

Response:

Thank you for your remark. As the Reviewer suggested, we have checked the figure 1.

The dot pattern results from the setting of a threshold. The assesment of EPCs was performed according to the standard procedure, as previously  described and published (Rohne P. et al. Journal of Clinical Medicine 2019; 8(11): 1984, 1-18; Rohne et al. Journal of Physiology and Pharmacology 2017, 68, 1, 139-148)

Thank You very much for all criticism. We believe that all of the above explanations and changes meet with Your approval and agreement. We hope that major revision of our paper made according to the Reviewer’s guidelines will improve our manuscript.

Reviewer 2 Report

In this manuscript, the authors assess the number of endothelial progenitor cells in the blood serum and their potential relationship with the clinical condition, radiological image and prognosis in patients in the acute phase of stroke caused by cerebral microangiopathy

Broad comments

Title:  The title is too long; it should be more concrete

Abstract: The abstract is unclear and does not reflect the content of the study

Methods:

  • Study type in not specified
  • Clarify how the control group participants were selected and their characteristics
  • Routine laboratory tests were performed within 24 hours from the onset of stroke symptoms. But what happens when the patients come to the hospital with a delay of more than 24 hours?
  • NIHSS and mRS scales must be referenced 
  • What is the reason why the authors choose the cutoff points of the NIHSS and mRS scale?
  • The authors have only specified the main variable of the study, but they have not indicated the rest of the variables that they will study and how they will be evaluated.

Results:

  • The results are written in a very confusingly and mesy way. The authors should re-write them
  • The authors use the word correlation when the statistical test they use is a different one
  • What is the association test used in the multivariate analysis? Especufy

Thank you very much

Author Response

Response to Reviewer 2 Comments

At the beginning, I would like to thank You for the careful review of our study and the constructive comments that have been used to organize all issues and improve the work.

Point 1:

The title is too long; it should be more concreto.

Response:

Thank You for this comment. According to the Reviewer’s suggestion we have shortened our title to be more concret ad concise (Lines 2-4).

Point 2:

The abstract is unclear and does not reflect the content of the study.

Response:

Thank You for this remark. We modified the abstract to be more clear  and transparent.

We have included the most important findings from our study.

Point 3:

Study type in not specified.

Response:

We are grateful to the Reviewer for the important opinion .We have added that this study is observational and prospective (Line 88).

Point 4:

Clarify how the control group participants were selected and their characteristics.

Response:

We extended information about control group (Lines 100-103). Some characteristics of control group subjects  can be also found in Table 1.

Point 5:

Routine laboratory tests were performed within 24 hours from the onset of stroke symptoms. But what happens when the patients come to the hospital with a delay of more than 24 hours?

Response:

We included only stroke subjects admitted to the hospital with duration of stroke symptoms no longer than 12 hours.  It was a deliberate action to standardize research results. According to the Reviewer’s kind remark we included this information in manuscript (Lines 99-100 and Lines 109-110).

Point 6:

NIHSS and mRS scales must be referenced 

Response:

Thank You for this suggestion. We referenced both scales (reference No 18 and 19).

Point 7:

What is the reason why the authors choose the cutoff points of the NIHSS and mRS scale?

Response:

We modeled on other relevant papers and general guidelines for evaluation of stroke severity. In American Heart/Stroke Asoociation guidelines- stroke with NIHSS more than 10 points is considered as dangerous for disability and  severe, with 3 or more points on mRS as a unfavorable. Thus, we adopted similar values in our study.

Point 8:

The authors have only specified the main variable of the study, but they have not indicated the rest of the variables that they will study and how they will be evaluated.

Response:

The main variable was the number of EPCs and all the statistical analysis was associated to this variable. Other variables were needed only to correlate with EPcs. Thus we described only main variable in detail, the rest of variables is described shortly but sufficiently (lines 111-121 and 158-166). Most of the data can be found in Tables. This study was large and analysis or specification of  other variables was not the point of interest of the research. We had to shorten the Method and Results Section to minimum. Manuscript’s capacity is limited and only most important content was included.

Point 9:

The results are written in a very confusingly and mesy way. The authors should re-write tchem.

Response:

Results Section style was  imposed  by another Reviewer, who asked us to apply the wording: „there were a significant” ; „there was no significant” to unify and systematize this section. We had to follow his remarks and this version was approved. As a result we have reduced our Results Section to minimum. We included only main information that could not be found in Tables. We are aware that a given style does not suit  everyone. Thus our Result Section is a compromise of many different suggestions and remarks. We modified it and did our best to improve this part. We subdivided it- one issue was raised in one subsection to make this part of manuscript more readible and transparent.

Point 10:

The authors use the word correlation when the statistical test they use is a different one

Response:

We appreciate this important comment. We have  corrected it and changed it according to the statistical test used for this purpose. At this moment the word „correlation” is used  only to show the results of Spearman’s  rank test.

Point 11:

What is the association test used in the multivariate analysis? Especify

Response:

We used regression- based asssociation test- standard multivariate analysis of variance (MANOVA).

We have made extensive changes of language and structure in our manuscript,  using the MPDI professional English editing service (ID english-18095).

Thank You very much for all criticism. We believe that all of the above explanations and changes meet with Your approval and agreement. We hope that major revision of our paper made according to the Reviewer’s guidelines will improve our manuscript.

Round 2

Reviewer 2 Report

The manuscript has improved considerably, but the authors should review the methodology and results section

Author Response

Response to Reviewer 2 Comments

At the beginning, I would like to thank You for the careful review of our study and the constructive comments that have been used to organize all issues and improve the work.

Point 1:

The manuscript has improved considerably, but the authors should review the methodology and results section

Response:

Thank You for this comment. We appreciate this important suggestion. According to the Reviewer’s  opinion we have  reviewed  the Methodology and Results Sections. We have made some language and style corrections in the manuscript (each revision is clearly highlighted using “Track Changes” function in MS Word). We did our best to to describe Methodology Section adequately and to present Results Section  clear and more transparent.

Thank You very much for all criticism. We believe that all of the above explanations and changes meet with Your approval and agreement. We hope that revision of our paper made according to the Reviewer’s guidelines will improve our manuscript.